# Kinetically controlled hetero-fusion is a systems-level behaviour of polymer nanoparticle populations

Stephen D. P. Fielden [1] ✉, Sean M. Collins [2,3], Matthew J. Derry [4], Caterina Ducati [5], Simon M. Fairclough [5], Alisha J. Miller[1], Rachel K. O'Reilly [1] & Paul D. Topham[4]

Particle fusion is key for establishing communication between biological components. For this reason, whole cell fusion plays a crucial role in many processes, including infection, muscle formation and tissue repair. Analogous co-assembly between synthetic nanoparticles represents a similar type of communication mechanism in artificial systems. Other approaches to control such co-assembly rely on incorporating anisotropic recognition units onto particle surfaces to provide a thermodynamic driving force. Here we present a fundamentally different approach, where hetero-fusion between two populations of undecorated polymer nanoparticles is regulated using kinetic control. Fusion extent is tuned simply by adjusting polymer chain length. Fusion is probed using an elemental tagging strategy for cryogenic scanning transmission electron microscopy combined with electron energy loss spectroscopy (cryo-STEM-EELS). Our results demonstrate the emergence of a complex process between populations of synthetic nanoparticles akin to communication. We anticipate such systems-level behaviour that results from hetero-fusion can enable future technologies.

Cell fusion[1,2] is tightly regulated to produce a specific hybrid and to ensure no damage occurs during membrane rearrangement[3–5]. This is seen upon fertilisation of an egg cell, which immediately triggers the cortical reaction to ensure no further fusion takes place[6]. Conversely, billions of cells are guided to fuse together to form the syncytiotrophoblast layer within the placenta[7]. Inspired by these intricate biological processes, chemists have developed methods to control the co-assembly of synthetic nanoparticles.

Early approaches to achieving this leveraged surface patchiness to control directional interactions between nanoparticles, in a manner reminiscent of molecular valance[8]. Solvophobic patches on different particles overlap, resulting in particle co-assembly. By combining two differently sized particles, hetero-assembly is thermodynamically favoured over homo-assembly, as the former produces a structure with lower solvent-exposed surface area[9]. The assembly of similar hierarchical structures[10] also can be induced by host-guest complexation[11], combining hard with soft particles[12], and temperature changes[13]. Other strategies, such as nucleobase pairing[14] can be used to produce hybrid nanoparticles[15]. Solvent evaporation[16] and polymerisation[17] can also be used to induce assembly of otherwise inert nanoparticles, in a manner more closely reminiscent of particle fusion in biology[18–20].

We wished to explore how to use kinetic control to regulate the degree of fusion between populations of different synthetic particles. Such emergent control of hetero-fusion would allow nanoparticles to act as building blocks for constructing complex hybrid materials and

[1]School of Chemistry, University of Birmingham, Edgbaston, Birmingham, UK. [2]School of Chemical and Process Engineering and School of Chemistry, University of Leeds, Leeds, UK. [3]Department of Materials, Imperial College London, London, UK. [4]Aston Institute for Membrane Excellence, Aston University, Aston Triangle, Birmingham, UK. [5]Department of Materials Science and Metallurgy, University of Cambridge, Cambridge, UK. ✉e-mail: s.fielden@bham.ac.uk

communication networks[21–23]. Here, we show a kinetically controlled mechanism of hetero-fusion mediated by polymer structure and length.

## Results

### Discovery of the hetero-fusion process

Hetero-fusion of polymer nanoparticles was achieved by leveraging ring-opening metathesis polymerisation-induced self-assembly (ROMPISA)[24–26]. This involves the extension of a hydrophilic P(norbornene) derivative formed from **G3** and either **NB-amine** or **NB-PEG** with hydrophobic **NB-MEG** in acidic aqueous solution (see Fig. 1 for chemical structures). The resulting amphiphilic copolymer chains self-assemble once the hydrophobic block reaches a critical length to produce spherical nanoparticles[27–29]. Subsequent rearrangement of rigid assembled P(norbornene) chains is inhibited[30], meaning further polymerisation of **NB-MEG** causes chains to become strained as they cannot adopt their preferred conformation. This causes the nanoparticles to reside in a non-equilibrium state. Eventually, with continued polymerisation of **NB-MEG**, spherical particles possess sufficient free energy to overcome the activation barrier to homo-fusion. Particles thus combine to produce elongated cylinders[28]. Therefore, there is a range for the average degree of polymerisation (DP) of **NB-MEG** where nanoparticles remain unfused but retain excess free energy; in other words, particles are held in a kinetically trapped state[23].

A greater DP of **NB-MEG** can be achieved in particles possessing a positively charged (i.e., P(**NB-amine**) containing), rather than neutral (i.e., P(**NB-PEG**) containing) outer surface before homo-fusion occurs. This is because electrostatic repulsion between positively charged particles and a higher corona water solubility generate a greater activation barrier to homo-fusion. Based on this, we hypothesised that combining kinetically trapped positively charged particles with neutral particles would result in hetero-fusion, because it would avoid interparticle charge repulsion and thus constitute a lower energy pathway to strain release.

In order to test our hypothesis, we synthesised two nanoparticle populations using ROMPISA at pH 2 (Fig. 1 and Supplementary Information, Section 2). The first population, **A200**, was formed of P(**NB-amine**)$_{11}$-block-P(**NB-MEG**)$_{200}$ chains, whilst the other population, **P100**, was formed of P(**NB-PEG**)$_{11}$-block-P(**NB-MEG**)$_{100}$ chains. All polymerisations occurred with complete consumption of each monomer as judged by proton nuclear magnetic resonance (¹H NMR, Supplementary Information, Supplementary Fig. 2–5). Resultant polymers exhibited low molar mass dispersity ($Đ < 1.2$), as determined by size exclusion chromatography (SEC, Supplementary Information, Supplementary Fig. 6, 7). Dynamic light scattering (DLS) and TEM (dry state and cryogenic) analyses showed both populations were formed principally (> 90% total particles) of spherical unfused nanoparticles with low size dispersity and similar diameters (hydrodynamic radius, $D_h$, = 36 nm, polydispersity [PD] = 0.12 for **A200**; $D_h$ = 33 nm, PD = 0.12 for **P100**. Dry state TEM particle length, $L_{TEM}$, = 29 ± 5 nm for **A200**; $L_{TEM}$ = 28 ± 5 nm for **P100**, Supplementary Information, Supplementary Fig. 8–10, 12, 14, 15). Dimensions determined by dry-state TEM were obtained by measuring 300 particles for each sample.

We then combined equal volumes of **A200** and **P100** dispersions to produce [**A200·P100**]. This resulted in an increase in turbidity within a few seconds, indicating the formation of larger particles. The increase of average particle size was evidenced by DLS ($D_h$ = 55 nm, PD = 0.17, Supplementary Information, Supplementary Fig. 17) and the formation of elongated fused particles was observed by dry state TEM ($L_{TEM}$ = 45 ± 23 nm, Fig. 1) and cryogenic (cryo-) TEM imaging (Fig. 1, Supplementary Information and Supplementary Fig. 23). Fusion was spontaneous upon combining **A200** and **P100**; no energetic input such as heating[31] was required to initiate the process. Histogram plots of the lengths of 300 particles each for **A200, P100** and [**A200·P100**]

samples (Fig. 2a) illustrate the formation of elongated particles upon hetero-fusion. As unfused **A200** and **P100** are very similar in size, the extent of fusion could be estimated by dividing the length of each measured particle in the [**A200·P100**] sample by the average unfused particle length (28.5 nm). This allows the number of unfused spherical particles present immediately prior to fusion to be determined (Supplementary Information, Section 4.1). This analysis indicated approximately 57% of spherical particles fused upon mixing (Supplementary Information, Supplementary Table 3) and that the majority of elongated particles (≈73% by number) consisted of two fused spherical particles.

To gain further insight into the fusion process, we analysed **A200, P100** and [**A200·P100**] dispersions using small-angle X-ray scattering (SAXS, Fig. 2b and Supplementary Information, Section 5). Unfused particles were modelled as spherical micelles and fused particles as cylindrical micelles[29,32]. The core diameter of **A200** and **P100** particles determined by SAXS (30 and 26 nm respectively) corresponded well with values obtained by TEM and DLS. Comparison of the scattering profiles further corroborated the formation of fused particles for [**A200·P100**]. A greater intensity at low $q$ and a shift in the major inflection to lower $q$ indicates particles for the [**A200·P100**] sample are, on average, larger. The feature around $q = 0.03 \text{ Å}^{-1}$ is less well defined in the [**A200·P100**] sample, indicating that the size of the particles was more polydisperse. The modelled cylinder length was short (33 nm), meaning the model likely also accounted for some of the larger unfused particles. Average aggregation numbers ($N_{agg}$) for **A200** and **P100** were determined to be 229 and 303, respectively. Fused cylinders had a $N_{agg}$ of 508 – a value close to the sum of $N_{agg}$ for **A200** and **P100** – further evidencing that the dominant fused species is formed from the hetero-fusion of two particles.

By using a synchrotron radiation source, the fusion of **A200** to **P100** could be monitored in real time[33,34]. As an increase in turbidity due to fusion was visually observed within seconds of mixing, we used a stopped-flow apparatus to combine particles and initiate fusion whilst irradiating with X-rays to facilitate time-resolved SAXS analysis. Fused particles formed this way were modelled satisfactorily using a single cylinder model, which showed an approximate logarithmic increase in $N_{agg}$ over the 300 s time period following mixing of **A200** and **P100** particles (Fig. 2c). The concomitant increase of particle size with increased scattering intensity provides further evidence for a greater degree of fusion over time. As the sample was not agitated during analysis, an upward turn in $N_{agg}$ after 300 s was likely due to damage from prolonged irradiation[35].

Knowledge of $N_{agg}$ values also permitted the number of particles per unit volume of dispersion to be determined, which in turn can be used to establish the number ratio of **A200** and **P100** particles upon mixing (Supplementary Information, Section 6). As synthesised, there are approximately 1.5 times more **P100** particles per mL of dispersion than **A200** particles. Therefore, combining equal *volumes* of particle dispersions gives a 3:2 *number* ratio of **P100** to **A200** particles prior to fusion. We adjusted this ratio by combining differing volumes of particle dispersions (Fig. 2d). An increasing extent of fusion manifests in a larger $D_h$ value[29]. The mixture possessing the highest $D_h$ value contained 70% **A200** dispersion by volume (corresponding to 60% **A200** particles by number or a 2:3 number ratio of **P100** to **A200** particles). TEM imaging of this mixture also revealed a greater extent of fusion: approximately 74% of spherical particles fuse upon mixing to give $L_{TEM}$ = 50 ± 26 nm (Supplementary Information, Supplementary Fig. 22, Supplementary Table 4). Conversely, a dispersion containing 70% **P100** dispersion by volume produced markedly less fusion: approximately 41% of spherical particles fuse to give $L_{TEM}$ = 38 ± 13 nm (Supplementary Information, Supplementary Table 5). Fusion extent is further reduced at more disparate particle ratios (Supplementary Information, Supplementary Tables 6 and 7). Fusion is suppressed by reducing the reaction temperature (Supplementary Information,

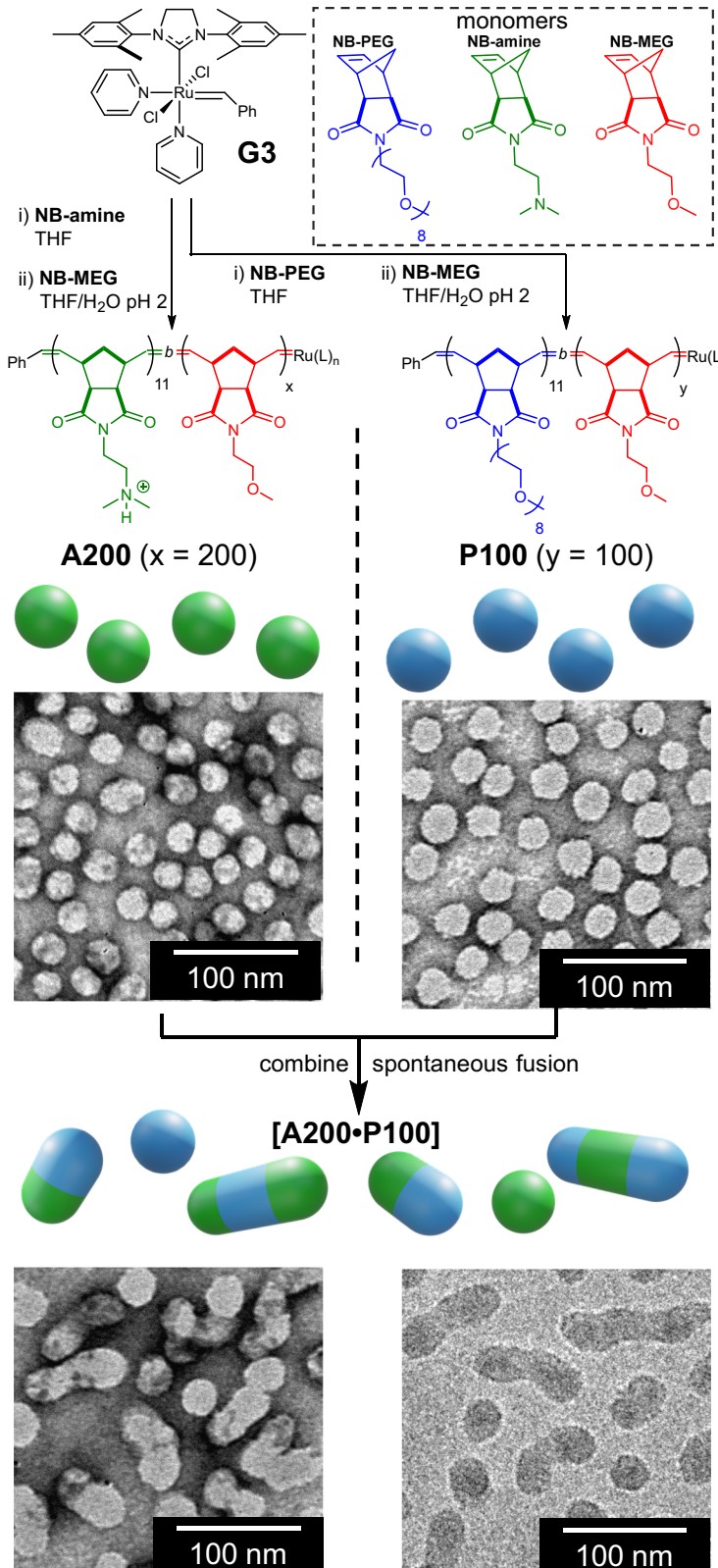

**Fig. 1 | Chemical structures and synthesis of polymer nanoparticles by ROM-PISA.** Nanoparticles are synthesised separately by using **G3** to polymerise **NB-amine** and **NB-MEG** to produce **A200** or **NB-PEG** and **NB-MEG** to produce **P100**. Combining **A200** and **P100** particles results in spontaneous fusion, as evidenced by dry-state and cryo-TEM. Samples analysed by dry-state TEM were stained with uranyl acetate (1% w/v) prior to imaging.

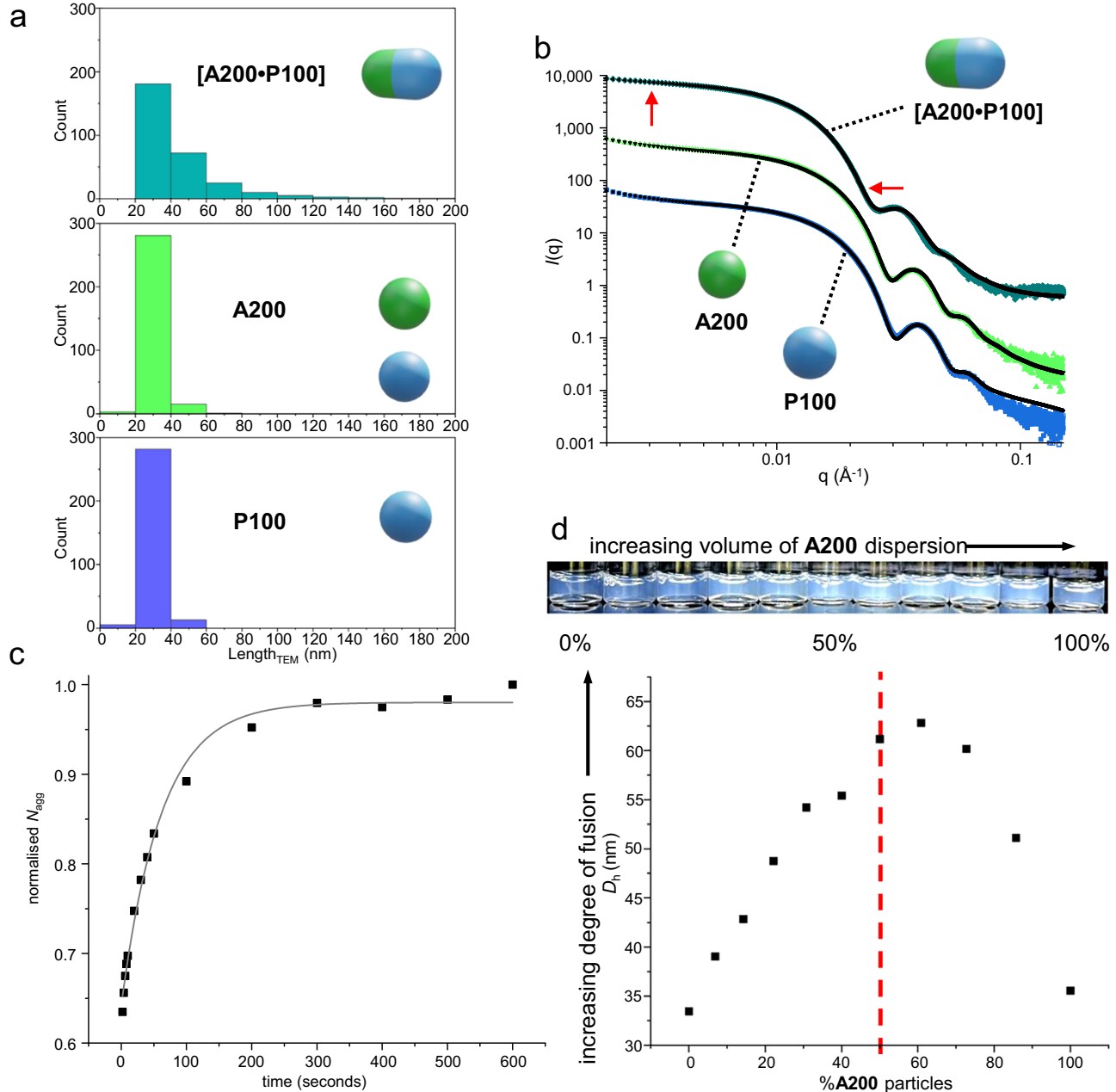

**Fig. 2 | Characterising hetero-fusion between A200 and P100 particles.**
**a** Histograms depicting number-weighted size distribution of fused and unfused particles, showing the formation of longer particles upon hetero-fusion. **b** Vertically stacked static SAXS data of unfused and fused particles. Blue scattering data is for **P100**, green for **A200** and aquamarine for **[A200•P100]** particles. Fits are shown in black. Red arrows indicate diagnostic changes in the SAXS pattern upon fusion. **c** Evolution of average $N_{agg}$ for fused cylindrical particles over time, as determined by in situ SAXS. Grey fitted line to guide the eye. **d** Determination of the optimum ratio of **A200** and **P100** particles to maximise fusion extent, as evidenced visually (shown with photos) and by average $D_h$. The red dashed line indicates a 1:1 number ratio of particles.

Supplementary Fig. 55) or is enhanced by using a greater proportion of THF (Supplementary Information, Supplementary Fig. 56).

These observations indicate the following regarding the fusion mechanism:

(1) **A200** and **P100** particles fuse with each other to produce cylinders that are formed of each starting particle. The alternative, whereby one particle population facilitates the homo-fusion of another, does not occur (Fig. 3a). If homo-fusion was the dominant mechanism, mixtures containing a significant excess of either particle would display the greatest extent of fusion.

(2) Particles containing longer P(**NB-MEG**) chains reside in a higher energy state prior to fusion[29]. Conversely, decreasing the P(**NB-MEG**) chain length suppresses fusion (Supplementary Information, Supplementary Fig. 57). Therefore, fused particles derived from the same number of spherical particles but with a higher average **NB-MEG** DP are more likely to form. This is likely why the presence of a slight excess of **A200** particles produces more fusion. This can be explained by considering the formation of fused cylinders formed of three spherical particles. These hybrid particles can be formed from either 2 × **A200** and 1 × **P100** particles (average DP = 160) or 1 × **A200** and 2 × **P100** particles

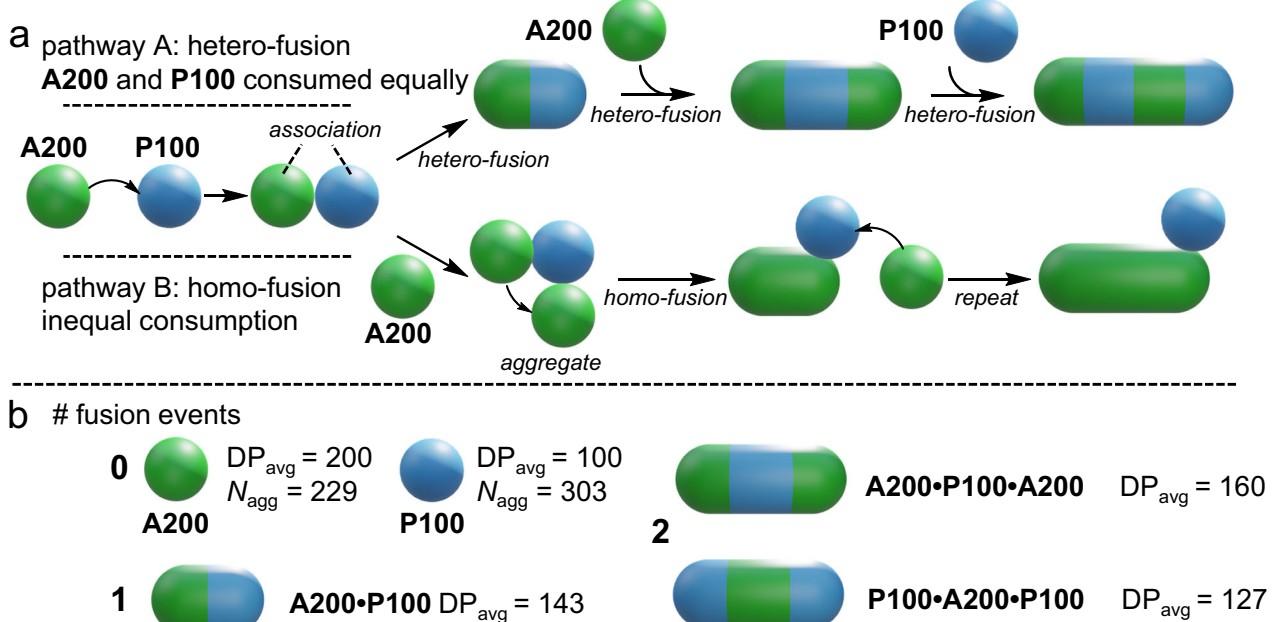

Fig. 3 | **Mechanism of hetero-fusion between A200 and P100 particles. a** Two potential pathways for the formation of elongated particles are possible. Pathway **a** is operative in this study, as evidenced by the need for a near 1:1 number ratio of particles to produce the maximum extent of fusion. **b** The average DP (DP$_{avg}$) of **NB**-**MEG** chains varies in different particle lengths and compositions. A higher average DP generates a greater driving force for fusion, meaning particles with a larger DP$_{avg}$ are more likely to form.

(average DP = 127). The former type is more likely to form due to its higher average DP, and so a slight excess of **A200** particles produces a greater extent of fusion (Fig. 3b). This is observed: 5% of particles are trimers upon fusion of the mixture containing a slight excess of **P100** (60% of starting particles by number), whilst 14% of particles are trimers in the mixture containing a similar excess of **A200** (61% of starting particles by number).

(3) Particles fuse by a nanoscale step-growth mechanism[36,37]. This can be deduced as the average extent of fusion per particle increases as greater numbers of fused cylinders are produced. Data from TEM images of fused samples correlate with the Carothers equation (Supplementary Information, Section 4.2).

**Multistep fusion**

Next, we investigated the effect on hetero-fusion of increasing **NB**-**MEG** DP in unmixed particles. We synthesised **A300**, P(**NB**-**amine**)$_{11}$-*block*-P(**NB**-**MEG**)$_{300}$, and **P150**, P(**NB**-**PEG**)$_{11}$-*block*-P(**NB**-**MEG**)$_{150}$. As expected, TEM imaging confirmed these particles had undergone significant homo-fusion during synthesis (49% spherical particles fuse for **A300** and 37% fuse for **P150**, see Supplementary Information, Supplementary Figs. 11 and 13 and Supplementary Tables 8 and 9) and were both longer and wider than **A200** and **P100**[28]. Notably, these larger particles were similar in length and width to each other (**A300**: $L_{TEM}$ = 44 ± 14 nm, dry state TEM particle width, $W_{TEM}$, = 32 ± 3 nm; **P150**: $L_{TEM}$ = 43 ± 13 nm, $W_{TEM}$ = 33 ± 4 nm).

Dispersions of **A300** or **A200** particles were combined in a 1:1 volume ratio with **P100** or **P150** particles to give three more mixed dispersions ([**A300·P100**], [**A300·P150**] and [**A200·P150**], Supplementary Information, Section 7). In all cases, $D_h$ values for these mixed dispersions were greater than those for individual populations (Fig. 4a and Supplementary Information, Supplementary Table 18), evidencing hetero-fusion within all mixtures. Therefore, hetero-fusion was evidently still possible with particle populations that had already undergone homo-fusion. For [**A300·P100**] and [**A200·P150**] mixed dispersions, the extent of hetero-fusion could not be precisely determined due to ambiguity arising from homo-fusion of **A300** or **P150**

and the consequent difference in average particle sizes for the two unfused populations. However, particles longer than the maximum observed in unmixed dispersions were present in [**A300·P100**] and [**A200·P150**] samples. Note that hetero-fusion also will have occurred to produce shorter cylinders, but these cannot be distinguished from those already formed by homo-fusion.

Comparing the ratio between the $D_h$ value for each mixed dispersion and the average for the corresponding unmixed populations allowed the extent of fusion to be qualitatively compared. These ratios were: 1.6 for [**A200·P100**], 2.1 for [**A300·P100**], 2.2 for [**A200·P150**] and 5.1 for [**A300·P150**]. Clearly, hetero-fusion was most extensive for the [**A300·P150**] mixture. Indeed, some fused particles were so long (>250 nm by cryo-TEM) they appeared tangled when imaged by dry-state TEM (Fig. 4b). Measuring non-tangled particles by dry state TEM indicated that around 90% of particles fuse on mixing ($L_{TEM}$ = 120 ± 76 nm, Supplementary Information, Supplementary Table 10); this is likely an underestimation given that only longer fused particles can tangle. SAXS analysis of the [**A300·P150**] sample indicated the average cylinder length in solution to be 156 nm (Fig. 4c and Supplementary Information, Supplementary Table 14). This further confirms the step-growth type mechanism of fusion, as long fused particles are only observed with high extents of fusion. In other words, the greater $L_{TEM}$ of [**A300·P150**] compared to [**A200·P100**] is due to the presence of more fusogenic starting particles.

Evidently, particles become more fusogenic as **NB**-**MEG** DP increases. However, this means particles that undergo a high extent of hetero-fusion also experience significant homo-fusion. Therefore, long hybrid particles can only be formed by a multistep fusion pathway, whereby particle populations experience homo-fusion first, followed by hetero-fusion (Fig. 4d)[38]. This means the [**A300·P150**] sample will contain a mixture of hybrid particles that are formed from either equal numbers of PEG and amine particles, as well as ones that are enriched in either type. This contrasts with the [**A300·P100**] and the [**A200·P150**], where fused particles will be, on average, enriched in amine or PEG chains, respectively. Thus, the fusion mechanism can be

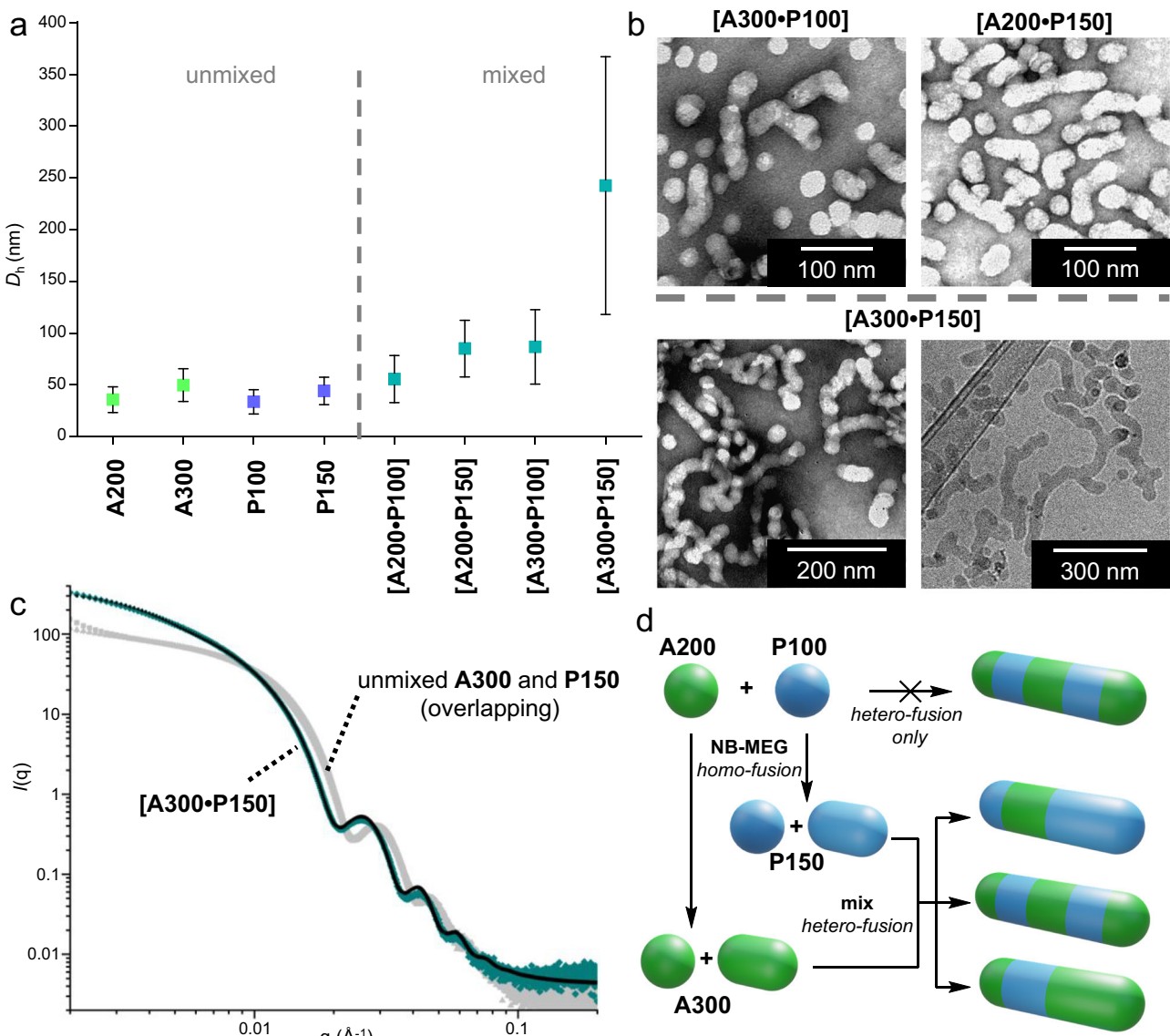

**Fig. 4 | The effect of varying NB-MEG DP in both positively charged and neutral particles on the extent of hetero-fusion. a** $D_h$ values increase as **NB-MEG** DP increases in both unmixed particles (homo-fusion) and when they are mixed (hetero-fusion). Error bars indicate standard deviation. **b** TEM images of fused mixtures, showing more extensive fusion. For the **[A300•P150]** mixture, extensive tangling of long fused particles is observed by dry-state TEM. Long fused particles are seen by cryo-TEM. Samples analysed by dry-state TEM were stained with uranyl acetate (1% w/v) prior to imaging. **c** Static SAXS data of **A300**, **P150** and **[A300•P150]** samples. The scattering data for **A300** and **P150** particles is in grey and overlaps as they have similar dimensions. Scattering data for **[A300•P150]** is coloured aquamarine; the fit is black. **d** It is not possible to produce long hybrid nanoparticles using hetero-fusion only; a multistep fusion sequence that combines both homo- and hetero-fusion is required. This produces hybrid particles that either are evenly formed of amine and PEG chains or are enriched in one type.

used to tune both the average length and composition of resultant hybrid nanoparticles.

## Determining the patterning of fused particles

There are two arrangements that polymer chains can adopt within hetero-fused particles. The first possibility is they are sufficiently mobile to diffuse throughout the entire particle to produce a uniform distribution. The other is they produce a striped surface pattern by overall remaining proximal to other chains originating from the same unfused particle (Fig. 5a). When imaged by cryo-TEM (Fig. 1 for **[A200•P100]**; Figs. 4b, 5b for **[A300•P150]**), the arrangement of polymer chains cannot be determined. This is partly because particle cores fuse to form a single continuous hydrophobic domain, rather than discrete segments[39]. This is why a plasticiser (10% tetrahydrofuran by volume) is required to provide sufficient chain mobility for fusion to

occur[40]. Tentative evidence of a striped pattern is gained from dry-state TEM imaging when uranyl acetate is employed as a negative stain. The stain accumulates on P(**NB-amine**) more than P(**NB-PEG**), possibly due to binding of the uranyl ion to amino groups[41]. This gives **A200** particles a darker appearance than **P100** particles (Fig. 1). Alternation of darker and lighter regions is observed on the surface of some **[A200•P100]** particles (Fig. 5c). However, stains often adhere to surfaces and particles unevenly and thus can produce misleading results[42]. We therefore sought a more reliable technique to determine the fate of polymer chains upon fusion. Our solution was to develop a method of elemental labelling suitable for electron microscopy. The surface patterning could then be determined when one unfused particle population is tagged (Fig. 5d).

Elemental mapping of soft matter within electron microscopy is challenging due to the ease of sample damage by the electron

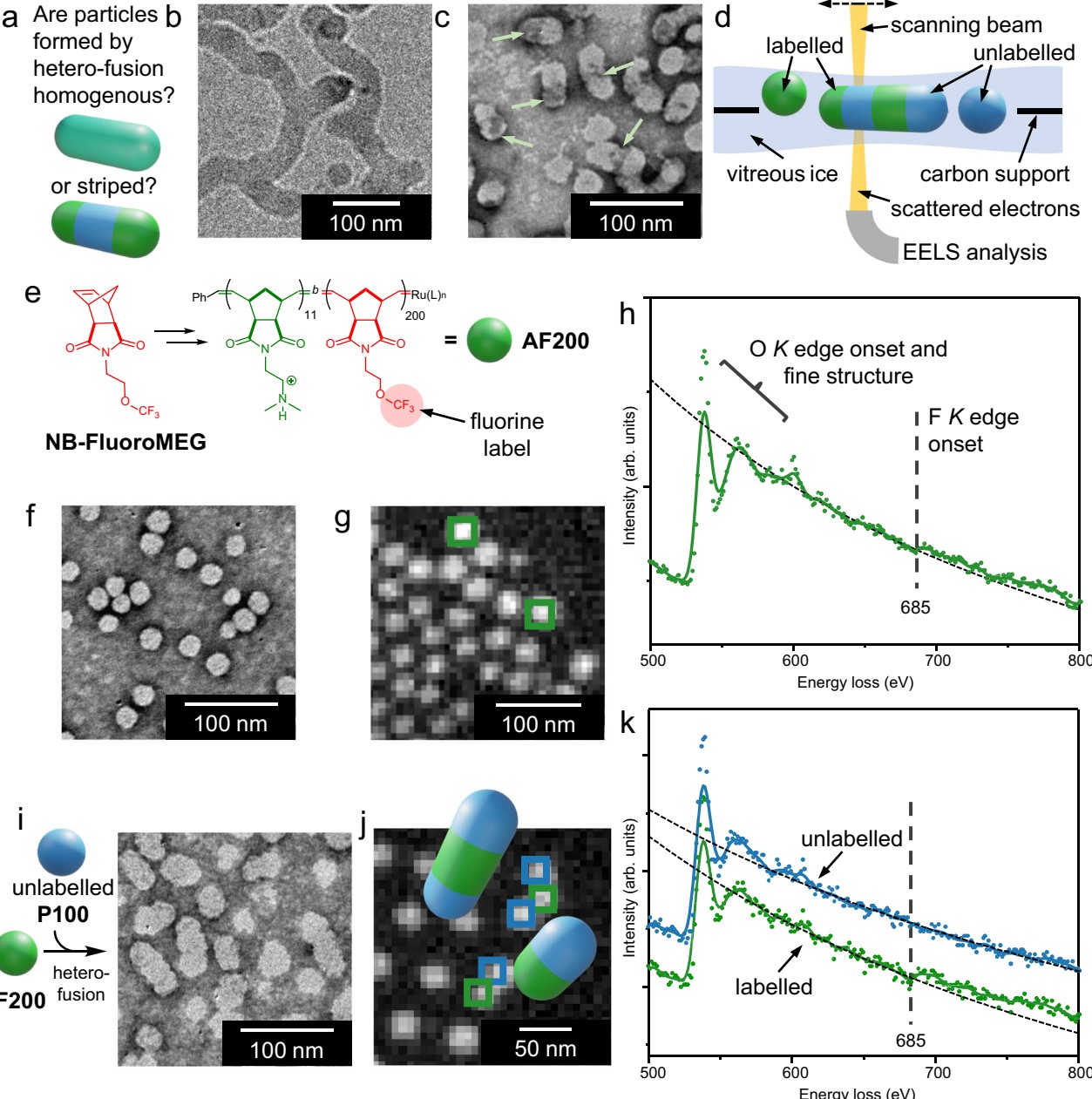

**Fig. 5 | Determining the patterning of particles formed by hetero-fusion. a** Two potential arrangements of polymer chains in fused particles. **b** Unlabelled **[A300•P150]** particles analysed by cryo-TEM appear uniform along their length. **c** Negative staining produces patterning (darker regions indicated by arrows) in some unlabelled **[A200•P100]** fused particles when imaged by dry-state TEM. **d** Analysis by STEM-EELS can reveal whether an elemental label is distributed uniformly or unevenly throughout particles. Cartoon components not to scale. **e** Synthesis of **AF200** from **NB-FluoroMEG** provides nanoparticles with a fluorine tag. **f** Dry-state TEM image of unmixed **AF200** particles. **g** Independent component analysis (ICA) carbon map of **AF200** particles under cryogenic conditions. **h** EELS spectrum for **AF200** under cryogenic conditions. The presence of fluorine is

evidenced by an onset of intensity at 685 eV. **i** Combining **AF200** and **P100** results in hetero-fusion, as evidenced by dry-state TEM imaging. **j** ICA carbon map of the **[AF200•P100]** sample. Regions analysed by EELS are indicated by boxes. Green boxes and plots indicate a region containing a fluorine signal, whilst blue boxes and plots indicate regions where no fluorine signal was observed. The most likely particle striping patterns are shown using these colours. **k** Summed EELS spectra for labelled and unlabelled regions. Note all regions were adjacent to another region with the opposite fluorine abundance (i.e., labelled regions were always adjacent to at least one unlabelled region). Samples analysed by dry-state TEM were stained with uranyl acetate (1% w/v) prior to imaging.

beam[43,44]. However, recent advances in scanning transmission electron microscopy (STEM) combined with electron energy loss spectroscopy (EELS) have given enhanced understanding of the structure and bonding within soft matter[45]. For example, the distribution of a nitrogen-rich active pharmaceutical ingredient encapsulated within polymer nanoparticles could be mapped using the relative intensity of the nitrogen $K$ edge[46]. Avoiding radiolytic damage induced by the

electron beam by minimising the electron fluence is key to this approach.

EELS offers both higher collection efficiency and superior light-element detection[47] compared to other electron beam spectroscopy methods, such as energy dispersive X-ray spectroscopy (EDS)[48]. Therefore, it offers the ideal opportunity to use common organic functional groups as tags for identifying polymer chains. The particles

discussed above that undergo hetero-fusion contain similar amounts of carbon, hydrogen, oxygen and nitrogen. Therefore, an additional element needed to be incorporated into one particle population to act as a tag. We selected fluorine, as it possesses a $K$ edge at a suitable energy of 685 eV[49] and because it is often found in organic molecules within the inert trifluoromethyl group[50]. To produce fluorine-tagged polymers, we performed ROMPISA with **NB-FluoroMEG** as the core-forming monomer using a slight modification to the aforementioned conditions (Fig. 5e and Supplementary Information, Sections 8 and 9) to access **AF200** particles formed of P(**NB-amine**)$_{11}$-*block*-P(**NB-FluoroMEG**)$_{200}$. Blocks of P(**NB-FluoroMEG**) contain 21% fluorine by mass. The similarity between **NB-FluoroMEG** and **NB-MEG** mean that **AF200** particles assembled comparably to **A200** particles (Fig. 5f), albeit they were slightly smaller ($L_{TEM} = 25 \pm 3$ nm). Some control over the ROMPISA process was lost, as a small number ( < 1% of the total, as determined by dry-state TEM) of notably larger particles are also formed.

Next, **AF200** particles were analysed by STEM-EELS under cryogenic conditions (i.e., in amorphous ice, so oxygen is present as $H_2O$) to determine whether fluorine was sufficiently abundant for its $K$ edge to be observed and to act as a control (Supplementary Information, Section 10). An independent component analysis[51–53] (Fig. 5g) was used to extract the carbon $K$ edge signal to map the location of particles on the grid (P(**NB-FluoroMEG**) contains 52% carbon by mass). Then, returning to the as-acquired data (i.e., without independent component analysis), EELS spectra for 14 particles were plotted (Supplementary Information, Supplementary Fig. 50) and summed (Fig. 5h). A clear onset of the fluorine $K$ edge was observed in > 60% of individual particle spectra. This gave rise to an enhanced signal in the summed data, demonstrating that EELS can be used to locate the position of polymer chains originating from **AF200**.

The hetero-fusion process was also tolerant to changes in monomer structure: combining **AF200** with **P100** produced [AF200•P100] (Fig. 5i). Particles in this sample had an average length greater than either unmixed population ($L_{TEM} = 38 \pm 12$ nm). The [AF200•P100] sample was similarly analysed by STEM-EELS and the location of particles again mapped by the carbon $K$ edge signal (Fig. 5j). Next, EELS spectra for 28 adjacent regions of fused particles were plotted (Supplementary Information, Supplementary Figs. 52 and 53) and summed (Fig. 5k). The fluorine $K$ edge signal was observed in the EELS spectra for approximately 50% of these regions. Notably, fluorine was not detected in adjacent regions. The abundance of fluorine instead alternated along fused particles, giving an indication of the most probable fluorine distribution within these particles. Particles located in different areas of the TEM grid showed the same behaviour. From this, it can be deduced that polymer chains do not distribute evenly throughout fused particles and instead form a striped pattern[8,9,17,39,54–56].

## Discussion

Based on the above results, we believe there are several criteria that must be met for hetero-fusion to be observed:

1. Nanoparticles must form in a thermodynamically unstable state (i.e., under kinetic control). This means a driving force is present for a morphological change to occur.
2. Slow or absent unimer rearrangement or exchange between particles, to prevent relaxation of the system by mechanisms other than fusion.
3. Fusion between different particles must have a lower energy pathway than fusion between the same particles to favour hetero-fusion.

In the system presented herein, rapid ROMPISA (full conversion in under 0.5 h) cause the polymerisation and assembly processes become

mismatched, meaning that polymer chains cannot adopt their most stable conformation, thus fulfilling criterion 1. The longer the chain, the further from thermodynamic equilibrium the system resides, presumably as more strain (i.e., excess free energy) is introduced within each particle core. Criterion 2 is satisfied using poly-norbornenes, as they are typically highly water-insoluble and reside in a glassy state at room temperature, meaning chain mobility is low. Previous research has shown that introducing protonated amine side chains on the surface causes particles formed by ROMPISA to resist homo-fusion due to inter-particle repulsion and corona solubility[28]. Therefore, criterion 3 is addressed by mixing **A200** particles with **P100** particles, as the latter do not have a positive surface charge. Preliminary results suggest that this is also possible at different pH values when using positively and negatively charged particles (Supplementary Information, Supplementary Fig. 58).

Whilst such kinetically controlled hetero-fusion has not been reported previously, a RAFT-PISA based system displays similar homo-fusion behaviour[57]. The use of a photo-RAFT protocol meant that polymerisation proceeded more quickly than a typical thermally-induced RAFT-PISA experiment ( > 99% monomer conversion in one hour). In addition, a graft-polymer based macro-chain transfer agent (CTA) was employed, meaning that multiple hydrophobic chains were covalently linked, likely severely limiting chain mobility. Thus, in this system, criteria 1 and 2 have been satisfied, and fusion was observed. An interesting experiment would be to combine particles formed from either an uncharged or charged macro-CTA to see if controlled hetero-fusion was possible.

Identifying and manipulating behaviour that results from combining interacting chemical species is a core focus of the nascent field of Systems Chemistry[58]. Broadening this strategy to include macro-molecules and nanoparticles is a grand challenge[59], but one that offers much-needed routes for the development of sophisticated materials[23,25,60–62]. One way that chemists have sought to expand complexity in nanoscale organisation is to develop methods to kinetically control the assembly of small-molecule building blocks into self-sorted structures[63]. Here, we present a fundamentally different approach, whereby pre-assembled nanoparticle building blocks display kinetically controlled 'social self-sorting' behaviour when combined. We have showed that, in contrast to other studies[8,9,11,13,17,19,36,37,39,56], patchy or anisotropic regions do not need to be engineered into nanoparticles to control their co-assembly pathway. Instead, hetero-fusion occurs in preference to homo-fusion because it possess a lower activation barrier. Such a process constitutes an elementary communication mechanism between synthetic nanoparticles reminiscent of cellular juxtacrine signalling[64]. Regulated fusion of compartments in biology is also important at the sub-cellular level, for example, with signalling mediated by synaptic vesicles[65], insulin secretion[66] and mast cell operation[67]. This hetero-fusion mechanism may find use in a number of potential applications. These include the targeted delivery of nanoparticles, whereby functionality held within different nanoparticles comes into close contact on demand. In turn, this could be useful for developing mechanisms of bifunctional catalysis[68], adaptive materials[69] and hierarchical assemblies[70]. Future work will exploit nanoparticle fusion to fabricate synthetic materials that display smart and responsive behaviours.

## Methods

• Synthesis of **A200, A300, P100** and **P150** by ROMPISA

A solution of **NB-amine** (11.6 mg, 50 µmol, 11 eq.) or **NB-PEG** (26.3 mg, 50 µmol, 11 eq.) in 900 µL filtered THF was rapidly added to a solution of **G3** (3.3 mg, 4.5 µmol, 1.0 eq.) in 100 µL filtered THF, contained within a 2 mL glass vial equipped with a stirrer bar. The resulting solutions were stirred rapidly for five minutes.

An aliquot (50 µL for **A200**, 33 µL for **A300**, 100 µL for **P100** or 67 µL for **P150**) of macroinitiator solution in THF was dispensed into a 2 mL glass vial containing a stirrer bar. Filtered THF was added to give 100 µL total volume in each vial. A solution of **NB-MEG** (10 mg, 45 µmol) in 0.9 mL of acidic phosphate buffer (pH = 2, PB2, final solids concentration = 1 wt%) was added rapidly to each vial. The resulting solution was thoroughly mixed by drawing up the entire volume into the pipette tip and ejecting the liquid back into the vial three times. The ROMPISA polymerisations were stirred at 300 rpm for 30 min to give **A200, A300, P100** and **P150** nanoparticle populations.

- Mixing **A200** and **P100** dispersions to produce **[A200•P100]**

An aliquot of **A200** was added to a 2 mL glass vial by pipette. A second aliquot of **P100** was added into the vial to produce a final volume of 200 µL. The resulting mixture was thoroughly mixed by drawing up the entire volume into the pipette tip and ejecting the liquid back into the vial three times. The mixture was left for 30 min without stirring before analysis.

- Cryo-STEM-EELS analysis of **AF200** and **[AF200•P100]** particles

### Experiment description

A sample of **AF200** or **[AF200•P100]** was diluted 100 × with PB2. Next, 5 µL of diluted sample was deposited onto a lacey carbon grid, followed by blotting for approximately five seconds. The grid was then plunged into a pool of liquid ethane, cooled using liquid nitrogen, to vitrify the sample using a ThermoFisher Vitrobot. Transfer into a pre-cooled cryo-TEM holder (Gatan 626 Cryo Holder) was performed under liquid nitrogen temperatures prior to microscopic analysis. The sample was transferred into a Thermo Fisher Scientific Spectra 300 microscope (University of Cambridge) equipped with a high brightness 'X-FEG' electron source and operated at 300 keV electron beam energy. The beam convergence semi-angle was set to 0.5 mrad. STEM imaging was carried out using the Panther detection system for simultaneous angle-resolved dark field and bright field image acquisition. Electron energy loss spectroscopy (EELS) data was acquired using a Gatan Continuum HR spectrometer, operated with a dispersion of 0.3 eV/channel. Regions suitable for analysis were identified using a combination of bright-field and annular dark-field imaging. This analysis also confirmed that morphological changes are not introduced during electron beam exposure. Cryo-STEM-EELS spectrum images (i.e., EEL spectra acquired at every probe position in a scanned field of view) were acquired for the selected regions with a spatial resolution (pixel size) of 7 nm. Sub-pixel scanning of the pixel area during EELS acquisition was implemented to spread the fluence across the entire pixel area. The parameters provided sufficient spatial resolution to resolve the nanoparticles while minimising electron beam-induced damage.

### Data analysis

To further control analyses, EELS measurements to evaluate fusion in **[AF200•P100]** particles were carried out over >10 fused regions across multiple separate areas where particles were observed within ice sufficiently thin for electron transparency (Supplementary Figs. 52, 53). The datasets were first aligned to remove any shifts in the spectrum on the camera by using cross-correlation based methods. This aligned the zero-loss peak throughout the spectrum image dataset to subpixel precision. Then, intensity artefacts arising from X-rays striking the detector camera were removed. This was achieved using routines in the HyperSpy 1.7.3 (https://zenodo.org/records/7263263) software package with interpolation with Poissonian noise after spike removal. The spectra were rebinned by a factor of 4 along the energy axis to reduce noise in the spectra.

Next, an independent component analysis (ICA) blind source separation algorithm was applied to the core-loss STEM-EELS data. The ICA decomposition consistently retrieved three components in these datasets: (1) One component showed signals at the C K edge, N K edge, and F K edge; (2)-(3) the other two components showed (2) O K edge intensity and (3) varying background. The C, N, and F containing component corresponded to localised intensity within the nanoparticles. This was selected as an overview map of the polymer contribution. Similar approaches have been successfully applied to separate chemical phases and background features in core-loss EELS[51–53]. Here, we use this map only as a map of the polymer distribution rather than seeking to directly interpret the spectra factors recovered by ICA. This overview map allowed all nanoparticles, labelled or unlabelled, to be located. This then provides a basis for identifying regions of interest (ROIs) for determining fluorine distribution. Selected area spectra were extracted with power law background fitting applied across a 'pre-edge' energy window at ~ 610–680 eV energy loss. The EELS data were then plotted for selected ROIs.

## Data availability

The data (SEC, SAXS, TEM, DLS, NMR, EELS) generated in this study are available in the Supplementary Information and have been deposited in the Mendeley repository under accession code https://doi.org/10.17632/jwkgr5xkn4.1. Data is also available from the corresponding author on request.

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

## Acknowledgements

S.D.P.F. is grateful for a Leverhulme Early Career Fellowship (ECF-2021-240, S.D.P.F.) and a Royal Society Dorothy Hodgkin Fellowship (DHF \R1\241133, S.D.P.F.), and to the University of Birmingham for funding. This work was carried out with the support of Diamond Light Source, instrument I22 (proposal SM33098, S.D.P.F., M.J.D., A.J.M., R.K.O.R. and P.D.T.), and we would like to specifically acknowledge the support received by Prof. Nick Terrill, Dr Andy Smith, and Dr Paul Wady during our experiment. The Aston Institute for Membrane Excellence (AIME) is funded by UKRI's Research England as part of their Expanding Excellence in England (E3) fund. S.M.C. acknowledges support from the EPSRC (EP/X040992/1, S.M.C.). We acknowledge the support of the Wolfson Electron Microscopy Suite and the Thermo Fisher Spectra 300 TEM funded by EPSRC (EP/R008779/1, C.D.). This work was also supported by the Henry Royce Institute for advanced materials through the Equipment Access Scheme, enabling access to the in-situ TEM at Cambridge; Cambridge Royce facilities grant EP/P024947/1 and Sir Henry Royce Institute - recurrent grant EP/R00661X/1.

## Author contributions

S.D.P.F. conceived the study. S.D.P.F. synthesised the polymer nanoparticles and performed SEC, DLS and dry-state TEM analyses. S.D.P.F. and A.J.M. synthesised monomers. S.D.P.F. and A.J.M. performed bright-field cryo-TEM analysis. S.D.P.F., M.J.D., A.J.M. and P.D.T. performed SAXS analysis. M.J.D. modelled SAXS data. C.D and S.M.F. planned and designed cryo-EELS experiments. S.D.P.F., S.M.C. and S.M.F. acquired EELS data. S.M.C. processed and analysed EELS data. R.K.O.R. provided laboratory space and support. S.D.P.F. wrote the manuscript. All authors discussed the results and edited the manuscript.

## Competing interests

The authors declare no competing interests.
