## [Transparent Peer Review file · Nature Communications]

Kinetically controlled hetero-fusion is a systems-level behaviour of polymer nanoparticle populations

Corresponding Author: Dr Stephen Fielden

Version 0:

Reviewer comments:

Reviewer #1

(Remarks to the Author)

Stephen and coworkers reported the hetero-fusion of two kinetically trapped polymer nanoparticles by ROMPISA. This work presents detailed discussion on the fusion process, however, the fusion mechanism is not clear for me. To make improvements, following lists my comments and suggestions.

1. What is the driving force of the fusion of A200 and P100 particles? The authors described that the fusion of A200 and P100 particles is kinetically controlled, without incorporating anisotropic interactions. However, from my point of view, the A200 NPs with amine shell are positively charged and P100 NPs with PEG shell are slightly negatively charged. The driving force of the fusion process is probably the electrostatic interaction, while the rigid assembled P(norbornene) chains makes this process under kinetically control. Hence, the authors are suggested to prepare some with low DP of PNB-MEG (such as A100 and P50) to ensure that these NPs are in thermodynamic steady state, and then investigate the mixing of A100 and P50 NPs to evaluate the fusion mechanism.
2. Temperature is a critical factor for regulating the rearrangement of core-forming chains. To prove whether the fusion process is under kinetically control, the fusion experiments at varied temperatures are encouraged. To meet the hypothesis from the paper, the fusion process may be greatly suppressed by lowering the temperature.
3. What about the heat change during the mixing of A200 and P100 dispersions? If the mixing is an exothermic process, the mixing process could bring the energetic input for particle fusion. The in situ DSC of the mixing process is suggested by the reviewer to monitor the heat change. Also the interactions between A200 and P100 particles could be identified.
4. Page 2, line 46. This is not only due to the electrostatic repulsion, the better solubility of PNB-amine than that of PNB-PEG is also a critical factor, that should be mentioned.

Reviewer #2

(Remarks to the Author)

See attached letter for detailed comments and revisions.

[Editorial Note: See end of file]

Reviewer #3

(Remarks to the Author)

This manuscript reports a new strategy to control hetero-fusion between block copolymer nanoparticles by manipulating surface charge properties and core block chain length. The authors further demonstrate, through STEM-EELS analysis, that fused particles exhibit nanoscale striped compositional patterns, providing important insight into the internal structure of the hybrids. The combination of fusion control and direct compositional visualization represents a significant technical advance. This work clearly brings novel conceptual and technical contributions to the field of polymer-based nanoparticle assembly. However, while the study introduces important advances, several critical issues remain. Several comments are provided below for the authors' consideration.

1. The interpretation of EELS data requires caution. The absence of a fluorine K-edge signal cannot be unequivocally interpreted as absence of fluorinated polymer, due to possible beam-induced degradation, spectral noise, and detection limits inherent to cryogenic STEM-EELS of soft matter. As an alternative or complementary approach, the authors could consider labeling the different nanoparticles with distinct fluorescent dyes and directly visualizing compositional patterns using super-resolution microscopy such as STED. This would provide an orthogonal validation of the observed hetero-fusion structures.
2. The authors have designed A200 and P100 particles with different NB-MEG block lengths, presumably to modulate the hydrophobic core stability and facilitate controlled fusion. However, the stability of the fused particles themselves is not investigated. It remains unclear whether the hetero-fused structures are kinetically trapped or thermodynamically stable. Assessing the long-term structural integrity of the fused nanoparticles (e.g., by time-dependent DLS, TEM, or storage stability studies) would strengthen the conclusions regarding the robustness of the fusion process.
3. The authors demonstrate hetero-fusion between positively charged and neutral particles based on minimization of electrostatic repulsion. However, by the same principle, hetero-fusion should also be possible between negatively charged and neutral particles. The manuscript does not discuss this possibility or provide experimental evidence addressing it. Including a discussion or experimental validation of anionic-neutral particle fusion would significantly strengthen the generality and conceptual robustness of the proposed fusion mechanism. If the authors have any relevant insights or preliminary data regarding anionic-neutral hetero-fusion, it would be valuable to include them.
4. In Figure 2b, the scattering profile of the fused particles exhibits a clear shift of the first minimum toward lower q values compared to the profiles of the individual unfused particles. This shift is consistent with an increase in particle size after fusion. However, the fitting parameters reported show little change relative to those of the unfused particles. This raises concerns about the validity of the fitting model used. The authors should carefully justify whether the chosen model appropriately captures the morphological changes upon fusion. If not, alternative fitting approaches should be considered.
5. The manuscript estimates aggregation numbers (N_{agg}) of the nanoparticles from SAXS scattering intensity at zero angle, $I(0)$. However, accurately determining $I(0)$ from SAXS measurements is challenging due to issues such as interparticle interactions, background subtraction errors, and instrumental fluctuations. As a result, the reported aggregation numbers—and therefore the derived particle number ratios used for mixing—may lack precision. To improve the accuracy of molecular weight and particle concentration determinations, complementary methods such as static light scattering (SLS) should be considered.
6. The introduction of fluorinated groups (NB-FluoroMEG) may alter the hydrophobicity and interparticle interactions compared to NB-MEG. The manuscript does not discuss whether this chemical modification could influence fusion behavior. A brief comment on the potential effects of fluorination would improve the interpretation of the results.
7. It appears that THF was used during the nanoparticle preparation and fusion processes. However, the manuscript does not clearly state whether the fusion behavior is robust to changes in solvent composition, such as variations in THF/water ratios or in the presence of different pH conditions. A brief discussion of how solvent conditions affect fusion efficiency and stability would enhance the generality and applicability of the conclusions.

Version 1:

Reviewer comments:

Reviewer #1

(Remarks to the Author)

The authors have largely addressed the reviewer comments and I think this work is now in a publishable form.

Reviewer #2

(Remarks to the Author)

I appreciate the thorough and thoughtful responses to my comments and suggestions. The authors have significantly strengthened their manuscript by adding new control experiments, clarifying the cryo-STEM-EELS methodology, and expanding the discussion on the generality and broader context of kinetically controlled hetero-fusion. The added statistical rigor and expanded data availability further enhance the scientific transparency and impact of the work. The consideration of additional fusion systems, temperature controls, and solvent effects provides helpful insight into the robustness of the findings. Overall, this revision satisfactorily addresses major concerns and makes a compelling case for the significance of programmable kinetic control in nanoparticle fusion. I support publication.

Reviewer #3

(Remarks to the Author)

The authors have thoroughly addressed all my previous comments. The revised manuscript is clear and technically sound. I

recommend acceptance as is.

REVIEWER COMMENTS

Reviewer #1 (Remarks to the Author):

Stephen and coworkers reported the hetero-fusion of two kinetically trapped polymer nanoparticles by ROMPISA. This work presents detailed discussion on the fusion process, however, the fusion mechanism is not clear for me. To make improvements, following lists my comments and suggestions.

Thank you for your comments, please find answers to your queries below.

1.1 What is the driving force of the fusion of A200 and P100 particles? The authors described that the fusion of A200 and P100 particles is kinetically controlled, without incorporating anisotropic interactions. However, from my point of view, the A200 NPs with amine shell are positively charged and P100 NPs with PEG shell are slightly negatively charged. The driving force of the fusion process is probably the electrostatic interaction, while the rigid assembled P(norbornene) chains makes this process under kinetically control. Hence, the authors are suggested to prepare some with low DP of PNB-MEG (such as A100 and P50) to ensure that these NPs are in thermodynamic steady state, and then investigate the mixing of A100 and P50 NPs to evaluate the fusion mechanism.

We agree that the mechanism is of profound importance, and we appreciate the considered suggestions on additional ways to interrogate it.

Rapid polymerisation during ROMPISA causes polymer chains to assemble in a kinetically trapped and unstable state (see ref. 29). Fused particles possess lower curvature and thus the hydrophobic core-forming block of polymer chains can adopt a more favourable conformation (a concept related to packing parameter). Hence fusion is a thermodynamically favoured process as it promotes the favourable rearrangement of chains within the particle hydrophobic core.

Previous work (refs. 28 and 29), as well as the occurrence of homo-fusion in this work, show that charge attraction between particle surfaces is not required for fusion to occur. Indeed, in ref. 29 we showed that re-introduction of a positive surface charge post fusion does not reverse the process. This indicates surface charge instead mediates the activation barrier to fusion. This is why hetero-fusion occurs selectively, as it avoids the close approach of like-charged particle surfaces.

Particle surface charges were determined from zeta potential measurements in previous work (ref. 28). Zeta potential values range from +20-30 mV for **AXXX** (XXX denotes a generic degree of polymerisation) particles and -0-3 mV for **PXXX** particles. Whilst there will be a small contribution from charge attraction in hetero-fusion, we note that a greater extent of fusion occurs when there is less charged polymer present in a given volume. For example, in a **[A200•P100]** formulation formed from a 1:1 volume ratio of **A200** and **P100** particles there is 0.57 mg of P(**NB-Amine**) and 2.6 mg of P(**NB-PEG**) per mL, whilst in a **[A300•P150]** formulation there is 0.38 mg of P(**NB-Amine**) and 1.8 mg of P(**NB-PEG**).

This is further illustrated by the experiment suggested by the referee. Combining **P50** (please note that these particles do not stain well due to their small size) and **A100** to give **[A100•P50]** results in no fusion (presented below and included in the revised Fig. S52). This demonstrates that despite the presence of charge attraction between the particles, there is insufficient driving force for fusion to occur. A reference to this experiment has been added to the revised manuscript, 'Conversely, decreasing the P(**NB-MEG**) chain length suppresses fusion

(Supplementary Information, Fig. S52.' and the figure has been added to the Supplementary Information under a new section 11: 'Additional control experiments.'

Figure S52 Attempted hetero-fusion of **A100** and **P50** particles. Particles stained with AGR1000 UA-Zero stain. Note **P50** particles do not stain well due to their small size

1.2 Temperature is a critical factor for regulating the rearrangement of core-forming chains. To prove whether the fusion process is under kinetically control, the fusion experiments at varied temperatures are encouraged. To meet the hypothesis from the paper, the fusion process may be greatly suppressed by lowering the temperature.

We agree that temperature should have an effect and so we carried out additional work to assess further the impact of this parameter. We conducted the standard fusion experiment between **A200** and **P100** particles at 0 °C by combining pre-cooled samples in an ice bath. As evident from the SAXS data, the hetero-fusion process is complete within five minutes at room temperature (rt). Therefore, to ascertain the effects of cooling, we quenched the cooled mixture after five minutes by adding buffer. This halts the fusion process by diluting THF, which inhibits chain mobility. TEM analysis (below and Fig. S50) of this experiment shows that, as the referee suggested, fusion is suppressed by lowering the temperature ($L_{\text{TEM}} = 31 \pm 5 \text{ nm @ } 0 \text{ °C}$; $45 \pm 23 \text{ nm @ rt}$). This provides further evidence for the kinetically controlled nature of the hetero-fusion process. We have added the statement, 'Fusion is suppressed by reducing the reaction temperature,' to the revised manuscript.

Figure: Attempted hetero-fusion of **A200** and **P100** at 0 °C. Particles stained with AGR1000 UA-Zero stain.

1.3 What about the heat change during the mixing of A200 and P100 dispersions? If the mixing is an exothermic process, the mixing process could bring the energetic input for particle fusion. The in situ DSC of the mixing process is suggested by the reviewer to monitor the heat change. Also the interactions between A200 and P100 particles could be identified.

We thank the reviewer for this suggestion. As discussed in point 1.1, only a very small amount of charged polymer is present. Particle dispersions are prepared with the polymer at 1% w/v. Therefore, almost all heat released upon mixing will be dissipated by the solvent. We do expect that charge attraction may be responsible for particle adhesion prior to fusion, but this is not a formal requirement for fusion to occur (see also point 1.1).

Regarding the fusion of particle cores, at this % polymer content, approximately 93 kJ/mol (based on monomer concentration) needs to be released by the polymer to heat the solvent by just 1 K (assuming it is pure water):

Concentration of monomer = 10 mg/mL = 0.0452 mmol/mL

Heat required to heat 1 mL of water = 4.2 J

Therefore, 4.2 J per 0.0452 mmol of (polymerised) monomer needs to be released to raise temperature by 1 K. This equates to 93 kJ/mol.

A typical value for such a fusion mechanism is no more than 25 kJ/mol (see *Proc. Natl. Acad. Sci. U.S.A.* **103**, 11916-11921 (2006) for an example with phospholipid membranes). As determined by the EELS experiments, complete mixing of polymer chains does not occur and thus the energy change and heating effect upon fusion is expected to be insignificant.

DSC will not show this information as it records the amount heat output at dynamic temperature. There are also technical difficulties, as fusion is too rapid to be accurately captured after being induced outside of the sample chamber. Therefore, whilst we carefully considered the idea, we do not believe it will reveal further insight.

1.4 Page 2, line 46. This is not only due to the electrostatic repulsion, the better solubility of PNB-amine than that of PNB-PEG is also a critical factor, that should be mentioned.

We thank the reviewer for this excellent point. We have added a clarifying comment in the revised manuscript, 'This is because electrostatic repulsion between positively charged particles and a higher corona water solubility generates a greater activation barrier to homo-fusion.'

Reviewer #2

Thank you for the opportunity to review your manuscript "Kinetically controlled hetero-fusion

is a systems-level behaviour of polymer nanoparticle populations.” This is an innovative study that presents a novel approach to controlling nanoparticle fusion through kinetic rather than thermodynamic considerations. Below are my detailed comments that I believe would strengthen this already impressive work.

We thank the reviewer for their comments, support, and enthusiasm for our work. We address their concerns below.

*Major Points

2.1 Cryo-STEM-EELS beam damage controls: The cryo-STEM-EELS analysis is a significant strength of your paper. While you mention using sub-pixel scanning to distribute electron fluence (Table S20), it would be valuable to explicitly discuss what controls were performed to confirm that the observed striped patterns are intrinsic to fusion rather than artifacts. For instance, did you observe consistent patterns across multiple particles and regions? Additional control experiments, such as imaging non-fused samples under identical conditions, would strengthen the reliability of your EELS data.

We wholeheartedly agree that carrying out careful controls is essential for cryo-STEM EELS analyses. In our revised text, we have sought to clarify the controls we have used including checking for consistent patterns across multiple fused **[AF200•P100]** particles and regions (Fig. S47 and S48) and non-fused **AF200** particles under identical conditions (Fig. S45 and S46).

We have amended the revised manuscript to emphasise these controls: ‘Next, **AF200** particles were analysed by STEM-EELS under cryogenic conditions (i.e., in amorphous ice, so oxygen is present as H₂O) to determine whether fluorine was sufficiently abundant for its K edge to be observed and to act as a control.’ And ‘A clear onset of the fluorine *K* edge was observed in >60% of individual particle spectra. This gave rise to an enhanced signal in the summed data, demonstrating that EELS can be used to locate the position of polymer chains originating from **AF200**.’

2.2 Generality across polymer systems: Your results are obtained with a specific ROMPISA system (norbornene-based amphiphiles in acidic water). To enhance impact, I suggest discussing whether such kinetically controlled hetero-fusion could occur in other polymer nanoparticle systems. What aspects of polymer chemistry or assembly conditions do you believe are critical for this behavior? Even a brief discussion of why chain length and kinetics might universally affect fusion (beyond this system) would help establish the broader relevance of your findings.

We have preliminary data that demonstrates fusion can occur in other norbornene-based systems, please see response 3.3.

We believe there are several criteria that must be met for this behaviour to be observed. The following text has been added to the conclusions of the revised manuscript:

‘Based on the above results, we believe there are several criteria that must be met for hetero-fusion to be observed:

1. Nanoparticles must form in a thermodynamically unstable state (i.e. under kinetic control). This means a driving force is present for a morphological change to occur.
2. Slow or absent unimer rearrangement or exchange between particles, to prevent relaxation of the system by mechanisms other than fusion.

3. Fusion between different particles must have a lower energy pathway than fusion between the same particles to favour hetero-fusion.

In the system presented herein, rapid ROMPISA (full conversion in under 0.5 h) cause the polymerisation and assembly processes become mismatched, meaning that polymer chains cannot adopt their most stable conformation, thus fulfilling criterion 1. The longer the chain, the further from thermodynamic equilibrium the system resides, presumably as more strain (i.e., excess free energy) is introduced within each particle core. Criterion 2 is satisfied using polynorbornenes, as they are typically highly water-insoluble and reside in a glassy state at room temperature, meaning chain mobility is low. Previous research has shown that introducing protonated amine side chains on the surface causes particles formed by ROMPISA to resist homo-fusion due to inter-particle repulsion and corona solubility.²⁸ Therefore, criterion 3 is addressed by mixing **A200** particles with **P100** particles, as the latter do not have a positive surface charge. Preliminary results suggest that this is also possible at different pH values when using positively and negatively charged particles (Supplementary Information, Fig. S53).

Whilst such kinetically controlled hetero-fusion has not been reported previously, a RAFT-PISA based system displays similar homo-fusion behaviour.⁵⁷ The use of a photo-RAFT protocol meant that polymerisation proceeded more quickly than a typical thermally-induced RAFT-PISA experiment (>99% monomer conversion in one hour). In addition, a graft-polymer based macro-chain transfer agent (CTA) was employed, meaning that multiple hydrophobic chains were covalently linked, likely severely limiting chain mobility. Thus, in this system, criteria 1 and 2 have been satisfied and fusion was observed. An interesting experiment would be to combine particles formed from either an uncharged or charged macro-CTA to see if controlled hetero-fusion was possible.'

2.3 Kinetic data and modeling: The time-resolved SAXS data (0–300 s) nicely captures the early fusion kinetics. It would be informative to know if longer-term measurements were performed to ensure the system reaches a stable endpoint. Additionally, you suggest a step growth- like fusion mechanism that correlates with the Carothers equation. Can you fit the kinetic data to an appropriate model (e.g., second-order kinetics) or provide a kinetic scheme? This would add rigor to your mechanistic understanding.

We thank the reviewer for recognising the value of the SAXS data.

The time-resolved SAXS data shows a clean increase in particle size for the first 300 seconds. After this time, the modelled N_{agg} plateaus (Fig. 2c), which we interpret to be the end point of the fusion process. This timeframe is in line with our other time-resolved SAXS studies of nanoparticle fusion (ref. 29). The nature of the time-resolved SAXS experiment means that the sample is exposed to high flux synchrotron X-ray radiation for 50% of the experiment time. This makes beam damage increasingly likely with passing time. We note that N_{agg} remains stable for at least 100 seconds after the 300 second time point. Therefore, we interpret the upturn in N_{agg} after this time as an artefact related to beam damage.

Ideally, we would have been able to fit the time-resolved data to a kinetic model. However, this data is noisy (see Fig. S32 and S33) because the sample displays low contrast and the collection time was necessarily short (20x less exposure time than for static samples) to capture time-resolved features. This means that data quality is not sufficient to model each nanoparticle population separately and instead a single cylindrical micelle model had to be used. This ambiguity means that, whilst qualitative understanding of the fusion process can be gained from the data, quantitative fitting is not possible. To illustrate this, see below for

attempted fitting of the first 50 seconds of N_{agg} values to a first order, $t \propto \ln(N_{agg})$, and second order $t \propto \frac{1}{N_{agg}}$ scheme.

Figure: Attempted fit to a first order scheme

Figure: Attempted fit to a second order scheme

2.4 Biological analogy and broader context: The introduction draws an analogy between this fusion process and biological juxtacrine signaling or cell-cell fusion. This connection could be strengthened by referencing a more concrete biological example (e.g., synaptic vesicle fusion). Additionally, discussing potential applications (targeted delivery, adaptive materials, hierarchical assembly) would help readers appreciate the importance of programmable nanoparticle fusion.

More concrete biological examples and a discussion around potential applications have been added to the conclusions of the revised text along with appropriate references:

'Regulated fusion of compartments in biology is also important at the sub-cellular level, for example with signalling mediated by synaptic vesicles,⁶⁵ insulin secretion⁶⁶ and mast cell operation.⁶⁷ This hetero-fusion mechanism may find use in a number of potential applications. These include the targeted delivery of nanoparticles, whereby functionality held within different nanoparticles comes into close contact on demand. In turn, this could be useful for developing new mechanisms of bifunctional catalysis,⁶⁸ adaptive materials⁶⁹ and hierarchical assemblies.⁷⁰

2.5 Statistical analysis: The quantitative results (e.g., percentage of fused particles) would benefit from an indication of variability. It is unclear if values like "57% fused" are from a single experiment or an average of several experiments. Including standard deviations and number

of replicates would strengthen the quantitative claims. If image analysis was used to count particles, stating the number of particles analyzed would enhance credibility.

We agree that transparent reporting of measurements and statistics is essential. The % fusion data was determined from a single batch of particles. We note particles cannot be stored and fused at a later date, so a batch of particles needs to be made each time a new experiment is performed. Repeating the experiment with a different batch of particles led to almost identical fusion results for the title reaction: **A200**, $L_{\text{TEM}} = 29 \pm 3$ nm; **P100**, $L_{\text{TEM}} = 30 \pm 4$ nm; **[A200•P100]** 1:1 volume ratio, $L_{\text{TEM}} = 43 \pm 14$ nm, 57% fused: 430 starting particles → 300 particles (184 remain unfused, 246 fused).

We have included the polydispersity index for all DLS data and standard deviations for all TEM measurements, as is normal practice for the field. DLS data was averaged from three measurements.

As indicated in the Supplementary Information, 300 particles were measured in all cases of image analysis to determine dimensions and fusion extent. This can also be seen by examining histograms in the Supplementary Information. A sentence explaining this has been added to the revised text.

*Minor Points

2.6 Figures and captions: Some figure captions could be more detailed. For example, in Fig. 5, clearly explaining the color coding in the EELS maps would help readers interpret which signals correspond to which elements/particles. Similarly, ensuring all axes in plots are clearly labeled with units would improve clarity.

We have revised the wording of figure captions to improve clarity. We have ensured that all axes in plots are labelled with units when necessary.

2.7 Long sentences in technical sections: Some sentences in the cryo-STEM analysis section are quite long and complex. Breaking these into shorter sentences would improve readability without sacrificing technical detail.

We have shortened sentences within the cryo-STEM analysis section for the revised text, methods and Supplementary Information.

2.8 Data availability: Ensuring that raw data (e.g., SAXS curves, TEM images, EELS spectra) are available in a public repository or as Supplementary Information would enhance reproducibility. A sentence confirming adherence to data-sharing policies would be appropriate.

We have now made data available in a public repository and added a data availability statement, as required by *Nature Communications* policy.

*Conclusion

This manuscript presents exciting research that advances our understanding of nanoparticle fusion mechanisms. The ability to control hetero-fusion through kinetic means rather than designed complementarity represents a significant paradigm shift. With the suggested revisions, this work will make an excellent contribution to *Nature Communications* and the broader field of nanomaterials.

We agree the ability to control nanoparticle fusion through kinetic control represents an exciting advance for nanomaterials.

We thank the reviewer for these detailed and positive comments, noting the significance of the contribution made in this work and its suitability for publication in *Nature Communications*.

Reviewer #3

This manuscript reports a new strategy to control hetero-fusion between block copolymer nanoparticles by manipulating surface charge properties and core block chain length. The authors further demonstrate, through STEM-EELS analysis, that fused particles exhibit nanoscale striped compositional patterns, providing important insight into the internal structure of the hybrids. The combination of fusion control and direct compositional visualization represents a significant technical advance. This work clearly brings novel conceptual and technical contributions to the field of polymer-based nanoparticle assembly. However, while the study introduces important advances, several critical issues remain. Several comments are provided below for the authors' consideration.

We agree that the ability to control fusion whilst also understanding the composition of our nanomaterials is significant advance. We thank the reviewer for their positive comments and address their concerns below.

3.1 The interpretation of EELS data requires caution. The absence of a fluorine K-edge signal cannot be unequivocally interpreted as absence of fluorinated polymer, due to possible beam-induced degradation, spectral noise, and detection limits inherent to cryogenic STEM-EELS of soft matter. As an alternative or complementary approach, the authors could consider labeling the different nanoparticles with distinct fluorescent dyes and directly visualizing compositional patterns using super-resolution microscopy such as STED. This would provide an orthogonal validation of the observed hetero-fusion structures.

We agree that the EELS data needs to be carefully interpreted. Based on this, we have re-phrased the text in the revised manuscript to emphasise that the particles shown in Fig. 5j display the *most probable* striping pattern based on the EELS data, rather than this being the *certain* structure.

In other words, the EELS experiments provide information of the relative likelihood of a certain region containing fluorine, as evidenced by the control experiment where unfused labelled particles were analysed (Fig. 5g, S45 and S46). In this control, we can be sure that all particles contained fluorine in the same abundance, because fluorine is incorporated into core-forming monomers before polymerisation. An onset of intensity at the fluorine *K* edge was not observed for all particles, but is clearly present when the signal from multiple particles is summed. We have clarified this by adding an additional figure into the Supplementary Information (Figure S46), which shows the signal-to-noise ratio improves as more labelled particles are analysed. These data confirm single particle detection is possible and that an unambiguous limit of detection in the spectra for ≥ 8 particles (i.e. fluorine *K* edge above background); the detection limit will be lower when integrating the intensity beyond the edge onset. Together, this analysis gives us confidence in the use of EELS to assess the striping pattern as the characteristic motif.

On the other hand, when considering the hetero-fused sample, summing particle regions next to fluorine-containing ones does not lead to a clear onset of intensity at 685 eV (Figure 5k). Striping is the only reasonable explanation for this, as this alternation of fluorine intensity is observed within particles from different areas of the TEM grid.

We also note in the text that the striping cannot be taken unequivocally as perfectly abrupt from the EELS analyses. The extent of chain mobility at the fusion interface is not known. Not all hetero-fused particles will have a perfectly alternating pattern due to homo-fusion during ROMPISA, as discussed in the text. For example, a small number of trimer particles will be derived partially from either homo-fused **AF200** or **P100** particles, so will have the structure **[AF200•AF200•P100]** or **[AF200•P100•P100]**.

We thank the reviewer for suggesting the use of a super-resolution technique, such as STED. Whilst we considered the use of a super-resolution technique, the use of expensive specialist dyes and access to bespoke microscopes would be required (i.e. collaboration with microscopy experts). The dyes would have to be further derivatized to form monomers suitable for ROMPISA. The concentration of dye present is an important parameter in super-resolution techniques, meaning it would likely require lengthy re-optimisation of the ROMPISA experiments. Finally, super-resolution techniques require the specimen to be fixed to a surface, which requires further experimentation and optimisation. We also foresee difficulty with fixation when using particles with different surface chemistries and dimensions, as well as potential interference with the fusion process.

Therefore, whilst we appreciate the suggestion, we feel that undertaking the other revisions suggested by all the reviewers have already significantly strengthened this work.

3.2 The authors have designed A200 and P100 particles with different NB-MEG block lengths, presumably to modulate the hydrophobic core stability and facilitate controlled fusion. However, the stability of the fused particles themselves is not investigated. It remains unclear whether the hetero-fused structures are kinetically trapped or thermodynamically stable. Assessing the long-term structural integrity of the fused nanoparticles (e.g., by time-dependent DLS, TEM, or storage stability studies) would strengthen the conclusions regarding the robustness of the fusion process.

During the course of this project, we found that the hetero-fusion needed to be performed and analysed on the same day that ROMPISA was performed to obtain reproducible results. We believe the reasons for this are twofold:

- (1) The active ruthenium species originating from **G3** cannot be quenched after the ROMPISA process, as this would involve adding ethyl vinyl ether, a known plasticizer for the resultant polymers. The active Ru therefore is able to catalyse cross-metathesis of polymer chains, which may allow chains to adopt a more stable conformation without the need for fusion.
- (2) Building on the point above, the particles are formed in a thermodynamically unfavourable state, so any chain mobility will eventually permit relaxation of the system.

These two processes also cause a slow morphological change to fused **[A200•P100]** particles, which takes a least several days to complete, versus five minutes for hetero-fusion. After two weeks, only spherical particles are observed by TEM (Fig. S49). However, the average particle diameter ($L_{\text{TEM}} = 35 \pm 6$ nm) is larger than for the starting **A200** and **P100** particles, indicating that fused particles likely collapse into a spherical shape over time. The hetero-fused structures are therefore kinetically trapped when first formed. We will investigate the mechanism that mediates this slow morphological change in future work.

Figure S49. [A200-P100] particles aged for two weeks. Particles stained with AGR1000 UA-Zero stain

3.3 The authors demonstrate hetero-fusion between positively charged and neutral particles based on minimization of electrostatic repulsion. However, by the same principle, hetero-fusion should also be possible between negatively charged and neutral particles. The manuscript does not discuss this possibility or provide experimental evidence addressing it. Including a discussion or experimental validation of anionic-neutral particle fusion would significantly strengthen the generality and conceptual robustness of the proposed fusion mechanism. If the authors have any relevant insights or preliminary data regarding anionic-neutral hetero-fusion, it would be valuable to include them.

We thank the reviewer for this incisive point. Please see reply 2.2 where we discuss the requirements for hetero-fusion are discussed.

We have preliminary data showing that fusion can occur in other norbornene based systems. We found **NB-COOH** works best as a corona-forming monomer to produce negatively charged particles at neutral pH. However, we also found that **NB-COOH** needs to be combined with **NB-PEG** to reproducibly form colloiddally stable particles. Therefore, negatively charged particles, **C200**, were made of the polymer P(**NB-PEG**)₁₁-*block*-P(**NB-COOH**)₅-*block*-P(**NB-MEG**)₂₀₀ at pH 7. These particles were spherical prior to hetero-fusion (Fig. S53).

Given **NB-PEG** was already situated in the negatively charged particles, it was most appropriate to study hetero-fusion to a positively charged particle population, rather than neutral (i.e., not **P100**).

A200 particles were colloiddally unstable above pH 4, so we made an additional population of positively charged particles, **D200**, that were stable at pH 7. These contained P(**NB-PEG**)₁₁-*block*-P(**NB-Amine**)₅-*block*-P(**NB-MEG**)₂₀₀ polymer. These particles were also spherical prior to hetero-fusion.

Combining **C200** and **D200** resulted in [**C200-D200**], clearly showing fusion also occurs in this system. We wish to emphasise the preliminary nature of this system, which has not been analysed by EELS or SAXS, but have included it in the revised Supplementary Information for completeness (Fig. S53). We have included the following statement in the revised manuscript: 'Preliminary results suggest that this is also possible at different pH values when using positively and negatively charged particles (Supplementary Information, Fig. S53).'

Figure S53. Hetero-fusion between **C200** and **D200** at neutral pH.

3.4 In Figure 2b, the scattering profile of the fused particles exhibits a clear shift of the first minimum toward lower q values compared to the profiles of the individual unfused particles. This shift is consistent with an increase in particle size after fusion. However, the fitting parameters reported show little change relative to those of the unfused particles. This raises concerns about the validity of the fitting model used. The authors should carefully justify whether the chosen model appropriately captures the morphological changes upon fusion. If not, alternative fitting approaches should be considered

The change in q value at the first minimum is subtle: **A200**, $q = 0.0298$; **P100**, $q = 0.0311$; **[A200•P100]**, $q = 0.0273$. These values correspond to an average core radius increase of approximately 2 nm upon fusion and an onset of anisotropy.

We originally attempted to fit the data for **[A200•P100]** to a spheres, dimers and trimers model (Fig. S28), but this gave an unsatisfactory result. In considering a range of models, we have confidence in the fitting carried out to the cylindrical micelle model (Fig. S27) as an appropriate method matched to the morphological changes observed.

3.5 The manuscript estimates aggregation numbers (N_{agg}) of the nanoparticles from SAXS scattering intensity at zero angle, $I(0)$. However, accurately determining $I(0)$ from SAXS measurements is challenging due to issues such as interparticle interactions, background subtraction errors, and instrumental fluctuations. As a result, the reported aggregation numbers—and therefore the derived particle number ratios used for mixing—may lack precision. To improve the accuracy of molecular weight and particle concentration determinations, complementary methods such as static light scattering (SLS) should be considered.

'The manuscript estimates aggregation numbers (N_{agg}) of the nanoparticles from SAXS scattering intensity at zero angle, $I(0)$.' - This is incorrect. N_{agg} is calculated by determining the volume of the particle core that is occupied by polymer (not solvent) and then dividing this by the volume occupied by a single core-forming chain. We refer the referee to equation S8 (for spheres) and equation S14 (for cylinders).

3.6 The introduction of fluorinated groups (NB-FluoroMEG) may alter the hydrophobicity and interparticle interactions compared to NB-MEG. The manuscript does not discuss whether this chemical modification could influence fusion behavior. A brief comment on the potential effects of fluorination would improve the interpretation of the results.

We agree that chemical changes may introduce multiple effects. We have considered these factors carefully also in altering the protocol used in the formation of **AF200** particles, as substituting a methoxy group for a trifluoromethoxy group increases a compound's hydrophobicity (*Molecules* **30**, 3009 (2025)). We have added a further comment in the revised text to more clearly outline this factor, 'To produce fluorine-tagged polymers, we performed ROMPISA with **NB-FluoroMEG** as the core-forming monomer using a slight modification to the aforementioned conditions (Fig. 5e and Supplementary Information, Sections 8 and 9)'.

Interparticle interactions are unlikely to be affected as only the hydrophobic core-forming polymer block is changed. We note that the fusion process operates similarly across both **A200** and **AF200** as evaluated by TEM observations.

3.7 It appears that THF was used during the nanoparticle preparation and fusion processes. However, the manuscript does not clearly state whether the fusion behavior is robust to changes in solvent composition, such as variations in THF/water ratios or in the presence of different pH conditions. A brief discussion of how solvent conditions affect fusion efficiency and stability would enhance the generality and applicability of the conclusions.

We thank the reviewer for raising this consideration. THF acts as a plasticiser in the fusion process. It can be regarded as a 'catalyst' for chain mobility. It also needs to be present to dissolve **G3** and hence form the corona-forming polymer prior to the addition of **NB-MEG** in buffer. Therefore, whilst THF cannot be removed from the process, more can be added.

We conducted the standard fusion experiment between **A200** and **P100** particles in the presence of 50% THF. This was achieved by adding additional THF immediately after mixing the particles. As can be seen from the TEM image below, this results in a greater extent of fusion ($L_{TEM} = 78 \pm 85$ nm @ 50% THF [note positive skew]; 45 ± 23 nm @ 10% THF). Evidently, more THF lowers the activation energy for fusion for a greater number of particles. At higher proportions of THF the polymer becomes soluble.

We have added the following statement to the revised manuscript: 'Fusion... is enhanced by using a greater proportion of THF (Supplementary Information, Fig. S51).'

Please see reply 3.3 regarding fusion at different pH conditions using alternative nanoparticles. **A200** particles are colloidal unstable above pH 4.

Figure S51. Hetero-fusion of A200 and P100 particles at 0 °C. Particles stained with AGR1000 UA-Zero stain.

Dear Authors,

Thank you for the opportunity to review your manuscript "Kinetically controlled hetero-fusion is a systems-level behaviour of polymer nanoparticle populations." This is an innovative study that presents a novel approach to controlling nanoparticle fusion through kinetic rather than thermodynamic considerations. Below are my detailed comments that I believe would strengthen this already impressive work.

*Major Points

1. **Cryo-STEM-EELS beam damage controls:** The cryo-STEM-EELS analysis is a significant strength of your paper. While you mention using sub-pixel scanning to distribute electron fluence (Table S20), it would be valuable to explicitly discuss what controls were performed to confirm that the observed striped patterns are intrinsic to fusion rather than artifacts. For instance, *did you observe consistent patterns across multiple particles and regions?* Additional control experiments, such as imaging non-fused samples under identical conditions, would strengthen the reliability of your EELS data.
2. **Generality across polymer systems:** Your results are obtained with a specific ROMP-ISA system (norbornene-based amphiphiles in acidic water). To enhance impact, I suggest discussing whether such kinetically controlled hetero-fusion could occur in other polymer nanoparticle systems. *What aspects of polymer chemistry or assembly conditions do you believe are critical for this behavior?* Even a brief discussion of why chain length and kinetics might universally affect fusion (beyond this system) would help establish the broader relevance of your findings.
3. **Kinetic data and modeling:** The time-resolved SAXS data (0–300 s) nicely captures the early fusion kinetics. It would be informative to know if longer-term measurements were performed to ensure the system reaches a stable endpoint. Additionally, you suggest a step-growth-like fusion mechanism that correlates with the Carothers equation. *Can you fit the kinetic data to an appropriate model (e.g., second-order kinetics) or provide a kinetic scheme?* This would add rigor to your mechanistic understanding.
4. **Biological analogy and broader context:** The introduction draws an analogy between this fusion process and biological juxtacrine signaling or cell-cell fusion. This connection could be strengthened by referencing a more concrete biological example (e.g., synaptic vesicle fusion). Additionally, discussing potential applications (targeted delivery, adaptive materials, hierarchical assembly) would help readers appreciate the importance of programmable nanoparticle fusion.

5. **Statistical analysis:** The quantitative results (e.g., percentage of fused particles) would benefit from an indication of variability. It is unclear if values like "57% fused" are from a single experiment or an average of several experiments. Including standard deviations and number of replicates would strengthen the quantitative claims. If image analysis was used to count particles, stating the number of particles analyzed would enhance credibility.

*Minor Points

1. **Figures and captions:** Some figure captions could be more detailed. For example, in Fig. 5, clearly explaining the color coding in the EELS maps would help readers interpret which signals correspond to which elements/particles. Similarly, ensuring all axes in plots are clearly labeled with units would improve clarity.
2. **Long sentences in technical sections:** Some sentences in the cryo-STEM analysis section are quite long and complex. Breaking these into shorter sentences would improve readability without sacrificing technical detail.
3. **Data availability:** Ensuring that raw data (e.g., SAXS curves, TEM images, EELS spectra) are available in a public repository or as Supplementary Information would enhance reproducibility. A sentence confirming adherence to data-sharing policies would be appropriate.

*Conclusion

This manuscript presents exciting research that advances our understanding of nanoparticle fusion mechanisms. The ability to control hetero-fusion through kinetic means rather than designed complementarity represents a significant paradigm shift. With the suggested revisions, this work will make an excellent contribution to Nature Communications and the broader field of nanomaterials.